# Effects of Lifestyle Intervention in Tissue-Specific Lipidomic Profile of Formerly Obese Mice

**DOI:** 10.3390/ijms22073694

**Published:** 2021-04-01

**Authors:** Norma Dahdah, Alba Gonzalez-Franquesa, Sara Samino, Pau Gama-Perez, Laura Herrero, José Carlos Perales, Oscar Yanes, Maria Del Mar Malagón, Pablo Miguel Garcia-Roves

**Affiliations:** 1Department of Physiological Sciences, Universitat de Barcelona, 08907 Barcelona, Spain; albagf@sund.ku.dk (A.G.-F.); p.gamaperez@gmail.com (P.G.-P.); jcperales@outlook.com (J.C.P.); 2Novo Nordisk Foundation Center for Basic Metabolic Research, University of Copenhagen, 2200 Copenhagen, Denmark; 3Universitat Rovira i Virgili, Department of Electronic Engineering & IISPV, 43004 Tarragona, Spain; ssamino@biosferteslab.com (S.S.); oscar.yanes@urv.cat (O.Y.); 4CIBER de Diabetes y Enfermedades Metabólicas Asociadas (CIBERDEM), Instituto de Salud Carlos III, 28029 Madrid, Spain; 5Department of Biochemistry and Physiology, School of Pharmacy and Food Sciences, Institut de Biomedicina de la Universitat de Barcelona (IBUB), Universitat de Barcelona, 08028 Barcelona, Spain; lherrero@ub.edu; 6Centro de Investigación Biomédica en Red Fisiopatología de la Obesidad y la Nutrición (CIBEROBN), Instituto de Salud Carlos III, 28029 Madrid, Spain; bc1mapom@uco.es; 7Nutrition, Metabolism and Gene Therapy Group, Diabetes and Metabolism Program, Institut d’Investigació Biomèdica de Bellvitge (IDIBELL), 08908 Barcelona, Spain; 8Department of Cell Biology, Physiology and Immunology, IMIBIC, Reina Sofía University Hospital, University of Córdoba, 14004 Cordoba, Spain

**Keywords:** lipidomics, tissue-specific, plasticity, energy intake, diet composition, exercise, hypothalamus, gastrocnemius, liver, white adipose tissue

## Abstract

Lipids are highly diverse in their composition, properties and distribution in different biological entities. We aim to establish the lipidomes of several insulin-sensitive tissues and to test their plasticity when divergent feeding regimens and lifestyles are imposed. Here, we report a proton nuclear magnetic resonance (^1^H-NMR) study of lipid abundance across 4 tissues of *C57Bl6J* male mice that includes the changes in the lipid profile after every lifestyle intervention. Every tissue analysed presented a specific lipid profile irrespective of interventions. Glycerolipids and fatty acids were most abundant in epididymal white adipose tissue (eWAT) followed by liver, whereas sterol lipids and phosphoglycerolipids were highly enriched in hypothalamus, and gastrocnemius had the lowest content in all lipid species compared to the other tissues. Both when subjected to a high-fat diet (HFD) and after a subsequent lifestyle intervention (INT), the lipidome of hypothalamus showed no changes. Gastrocnemius and liver revealed a pattern of increase in content in many lipid species after HFD followed by a regression to basal levels after INT, while eWAT lipidome was affected mainly by the fat composition of the administered diets and not their caloric density. Thus, the present study demonstrates a unique lipidome for each tissue modulated by caloric intake and dietary composition.

## 1. Introduction

The dramatic increase in the prevalence of obesity around the world is a major health concern due to its contribution to increased risk for several pathologies such as cardiovascular diseases, cancer and diabetes. In obese patients, decreased sensitivity to insulin in peripheral tissues, mainly skeletal muscle, liver and adipose tissue, is a major transitional hallmark before overt diabetes. The onset of insulin resistance in those tissues has been attributed to the failure of pancreatic β-cells to cope with the increased demand for insulin due to excessive nutritional intake leading to lipid accumulation [1]. Lipotoxicity starts with insulin resistance in adipose tissue that causes a failure in suppressing lipolysis and in the capacity of adipose tissue to take up lipids from the circulation and expand. Elevated circulating free fatty acids (FFA) levels lead to ectopic fat deposition [2]. Lipotoxicity has been associated with metabolic alterations in fatty acids (FA) utilisation and intracellular signalling [3]. Hence, in insulin-sensitive tissues, mitochondrial dysfunction is induced by the chain reaction that starts with lipid oversupply. Firstly, the machinery in the mitochondria responsible for fatty acid oxidation is overloaded, promoting an increase in reactive oxygen species (ROS) production [4]. Then, uncoupling of the oxidative phosphorylation follows and the permeabilisation of the outer mitochondrial membrane causes further alterations in mitochondrial structure and function [5]. Thus, there is a major interest in the interplay between lipid accumulation and mitochondrial dysfunction, which demands more knowledge on lipid profiling, that is, understanding their identity and distribution in each tissue upon various dietary regimens.

Lipids are a family of biological substances found across all living cells as hydrophobic or amphipathic macromolecules. They have evolved to become essential for various functions as structural components, signalling and regulatory molecules and energy storehouses, among others. In mammals, lipids can be either consumed from exogenous sources or synthesised de novo, and they are modified by various enzymes and other metabolites, creating a unique lipid profile or “lipidome”. Each individual lipid species has unique physical and chemical properties, allowing the classification of lipids based on fatty acyl chain lengths and degrees of saturation [6]. Lipids are classified into 8 main groups: fatty acyls, glycerolipids, glycerophospholipids, sphingolipids, sterol lipids, prenol lipids, saccharolipids and polyketides [7].

Lipid profiles vary within the layers of a membrane, within the cells of the same tissue, between tissues and between animal species. In addition to lipid diversity in spatial distribution, a temporal factor should be considered, where the circadian rhythm and the life cycle of an organism are examples of such effect [8]. Alterations in the lipidome can also be attributed to a pathological condition where lipid homeostasis is disrupted due to genetic disorders, cancers, cardiovascular diseases, obesity and diabetes, among others [8]. Furthermore, modifications to the lifestyle of an individual like changes in nutrition and physical activity can greatly affect a lipid profile. As a result of all the factors that could alter it, a lipidome can be described as a dynamic and flexible system, which makes its analysis challenging [8]. Even though lipidomics has advanced, there is still a gap in the lipid profiling of numerous tissues at different physiological states.

Our aim was to assess the lipidomes of insulin-sensitive tissues in an experimental model of obesity-related type II diabetes (T2D) through high-fat feeding in mice, and the changes evoked by a dual body weight-reducing intervention focusing both on nutrition (composition of the diet plus restriction in caloric intake) and physical activity. This work presents novel data regarding the abundance and distribution of lipids across various tissues, the specificity they exhibit for tissues critical in the aetiology of obesity-related T2D (liver, epididymal white adipose tissue (eWAT), gastrocnemius and hypothalamus) and the alterations induced by lifestyle changes.

## 2. Results

### 2.1. High-Fat Diet Induced Phenotype Is Reversed by Nutritional and Exercise Intervention

We investigated the effect of a 16-week high-fat diet (HFD) on several phenotypic parameters and its reversibility after a lifestyle intervention. Three experimental groups were included in the study: control group (Ctrl), HFD-induced obese group (HFD) and the intervention group (INT), which underwent a sequential high-fat feeding and a calorie restriction and diet substitution (from saturated fat and simple sugars to poly- and mono-unsaturated fat and complex carbohydrates) (Table 1), together with an exercise program (incremental treadmill training). The HFD group showed an increase in body weight (Figure 1A) as well as in normalised liver and eWAT tissue weight (Figure 1B). HFD animals also showed an increased glycemia AUC (area under the curve) on a glucose tolerance test after 14 weeks of high-fat feeding (Figure 1C), indicative of impaired glucose handling. The changes observed in response to HFD were all reverted after intervention. Thus, the INT group had a body weight (Figure 1A), liver and eWAT weights (Figure 1B) and a response to intraperitoneal glucose tolerance test (IGTT) similar to the Ctrl group.

Body weight and glucose handling differences among groups correlated well with oxygen consumption, heat dissipation and locomotor activity, as measured by indirect calorimetry. The HFD group consumed less oxygen than controls during the light and dark cycles, whereas the INT group showed no differences in oxygen consumption (Figure 1D). Heat dissipation was inferred from indirect calorimetry data using the Weir equation, indicating a decrease in heat production in HFD mice both in the light and dark phases as compared to Ctrl animals, whereas the INT group produced less heat in the light phase (Figure 1E). Locomotor activity both as ambulatory movement (active movement) and fine movement (grooming and in-place movement) showed no changes for HFD and INT groups in comparison with Ctrl (Figure 1F,G). However, HFD and INT groups showed marked differences (Figure 1F), with the INT group showing more ambulatory movement than the HFD group during the light phase. This extra activity might be explained by the fact that INT mice were under calorie restriction and their diurnal activity could be related to their seeking for food.

### 2.2. Lipid Profile Is Tissue-Specific

The primary aim of this study was to evaluate the lipid profile of insulin-sensitive tissues in our three experimental groups (Ctrl, HFD and INT). Due to the unique function and structure of each tissue and to the different properties of each lipid family, it was of interest to, firstly, illustrate the differential lipid profiles of tissues from lean healthy mice (Ctrl group) (Appendix A). In these animals, we could clearly identify some tissue restriction for specific lipids: (i) eWAT and liver are the tissues with the highest content of glycerolipids (Figure 2A–C), whereas (ii) eWAT presented the highest FA content (Figure 2H–O) and (iii) hypothalamus showed the highest content in sterol lipids, and phosphoglycerolipids (Figure 2E–G,P–V). Triglycerides (TAG) were present at higher levels in eWAT than in liver and gastrocnemius muscle (Figure 2A). Also, diglycerides (DAG) were higher in eWAT than in liver (Figure 2B), and monoglycerides (MAG) were mainly detected in liver (Figure 2C), and free glycerol was only detected in eWAT (Figure 2D). Moreover, sterol lipid species such as total cholesterol (Chol) and free cholesterol (FC) were higher in the hypothalamus as compared to the other 3 tissues (Figure 2E,F), and esterified cholesterol (CE) was only detected in hypothalamus and liver (Figure 2G). The peak in the spectra of the NMR representing total cholesterol includes both free cholesterol and esterified cholesterol. Then, two different peaks in the spectra allow the quantification of FC and EC independently. Fatty acid profile was also different for the 4 tissues: (i) unsaturated fatty acids (UFA) (Figure 2H), polyunsaturated FAs (PUFA) (Figure 2I), oleic acid (Figure 2J), arachidonic and eicosapentaenoic acids (ARA-EPA) (Figure 2N) and omega3- docosahexaenoic acid-EPA-Linolenic (n3.D-E-L) (Figure 2O) were highest in eWAT, with liver and hypothalamus in second place, and a very low content present in gastrocnemius muscle. (ii) Linolenic acid was only detected in eWAT (Figure 2L) while linoleic acid was found in all 4, with highest levels in eWAT (Figure 2K). Also, (iii) docosahexaenoic acid (DHA) was higher in liver and hypothalamus than in gastrocnemius, and not detected in eWAT (Figure 2M). Finally, phosphoglycerolipids were mostly restricted to hypothalamus, with total phospholipids (Total PL) only detected in this tissue (Figure 2V), and phosphocholine (ChoP), phosphatidylethanolamine (PE) and plasmalogen and sphingomyelin (SM) were mainly found in hypothalamus (Figure 2P,R,T,U), whereas sn-glycerophosphocholine (sn-GPC) was also found in liver in addition to the hypothalamus (Figure 2S).

Next, we assessed the lipid profile and tissue distribution changes associated to high-fat feeding and the nutritional and exercise intervention following the HFD.

### 2.3. HFD and Nutritional and Exercise Intervention Did Not Alter the Lipid Profile of Hypothalamus

Neither the levels of fatty acids (Figure 3A,B), sterols (Figure 3C) nor phospholipids and sphingomyelin (Figure 3D,E) in the hypothalamus were affected by HFD or diet + exercise (INT) interventions. The fact that the total content of the different lipid classes of the hypothalamus was not affected by the HFD nor the intervention that followed, including a change in dietary lipids, indicates that the hypothalamus is either not sensitive to those nutritional alterations or the modifications to the lipidome were not detected due to the limitations in the sensitivity of the NMR technique. The levels of fatty acids (Figure 3A,B), sterols (Figure 3C) phospholipids and sphingomyelin (Figure 3D,E) were not affected by either HFD or diet + exercise (INT) interventions. The fact that the lipid profile of the hypothalamus was not affected by the HFD nor the intervention that followed, which included a change in dietary lipids, indicates that the hypothalamus is not a depot for dietary lipids, probably as the blood–brain barrier serves as a metabolic shield to the brain. Nonetheless, through NMR, it is not possible to recognise the changes in the composition (level of unsaturation and total carbon length) of the lipid species amongst a lipid class.

### 2.4. Lipid Profile of Gastrocnemius Was Sensitive to Alterations in Energy Balance

We assessed the effect of HFD and the intervention on the lipid profile of the gastrocnemius skeletal muscle. We observed an increase of TAG (Figure 4A) and several fatty acids, such as combined UFA, oleic acid and linoleic acid (Figure 4B), in HFD-fed animals. No changes were observed in other fatty acids like PUFA (Figure 4B), DHA, ARA-EPA and n3.D-E-L (Figure 4C), or in sterol lipids (Figure 4D) and phospholipids (Figure 4E,F). When comparing the Ctrl group to the INT group, the content of TAG was identical, which means there was a regression to normal values after the effect of HFD (Figure 1A). The same reversibility was observed for fatty acids, UFA, oleic acid and linoleic acid (Figure 1B). On the other hand, the levels of ARA-EPA were decreased and the levels of n3.D-E-L increased in the INT as compared to the Ctrl group. There were no alterations in sterols (Figure 4D) and phospholipids (Figure 4E,F) in the profile of INT as compared to Ctrl. These results all together suggest that the lipid profile of the gastrocnemius is reflective of changes in energy intake, and do not mirror the composition of the administered diet (Table 1).

### 2.5. High-Fat Diet Induced Increase in Most Lipid Classes Content, and Its Reversibility after Intervention Was a Marker of the Liver Lipid Profile

Liver showed an increase in TAG and DAG levels after HFD (Figure 5A,B). Similarly, the contents of all fatty acids: UFA, PUFA, oleic acid, linoleic acid (Figure 5C), DHA, ARA-EPA and n3.D-E-L (Figure 5D), and Chol and CE levels (Figure 5E), were higher in the HFD group. Although there was a decrease of SM and PE, there wereno changes observed for ChoP, plasmalogen, sn-(GPC) and phosphatidylcholine (PC) (Figure 5F,G). The accumulation of TAG (Figure 5A), DAG (Figure 5B) and all the fatty acids but n3.D-E-L (Figure 5C,D) were recouped after the intervention phase. In the INT group, n3.D-E-L levels were increased compared to Ctrl and HFD (Figure 5D). The pattern of reversibility was also observed for Chol and CE contents (Figure 5E). Also, INT animals showed decreased levels of ChoP, plasmalogen and SM (Figure 5F) when compared to Ctrl. Overall, these results show that the lipid profile of the liver underwent major alterations under the stress of the HFD and were reversed upon intervention. This proves the liver as an important site for depot and lipid metabolism, especially after esterification of fatty acids from the diet, as would be expected. Therefore, these results suggest that the content of lipids in the liver mostly reflect the energy balance and body composition; after a high-fat feeding, the levels of most lipids are increased and recover after a lifestyle intervention that includes nutritional and energy constraints.

### 2.6. eWAT Lipid Content Mirrored the Composition of the Diet

The content of glycerolipids in eWAT was unaffected by HFD feeding and intervention, except for the increase in free glycerol observed in the INT group as compared to Ctrl (Figure 6A,B). However, fatty acids, such as linoleic acid and ARA-EPA, were decreased (Figure 6D,E). When comparing the INT group to Ctrl animals, we observed a decrease in the levels of linoleic acid and ARA-EPA and an increase in levels of oleic acid, PUFA, linolenic and n3.D-E-L (Figure 6C–E). The HFD group showed an increase of Chol and FC sterol lipids that persisted after intervention (Figure 6F). The changes mentioned above demonstrate that the lipid profile of eWAT is highly dependent upon the administered diet, reflecting the role of WAT in the storage of lipids. Both HFD and INT diets have higher fat content (45%) than Ctrl, but the INT diet contains less saturated fat and more unsaturated fat than the HFD (Table 2), which mirrored the changes observed in the levels of fatty acids as quantified by NMR (Figure 6C–E).

## 3. Discussion

Lipid profiles are subjected to changes, in a tissue-specific manner, related to the physiological state of the organism and to the environmental conditions. Lipid excess caused by chronic overnutrition, in most cases, leads to insulin resistance (IR) and mitochondrial dysfunction, which in turn induces further modifications of the lipidome. Therefore, it is clear that the tissue distribution and the profile of those lipids in many tissues can be greatly affected by nutritional interventions, changes in the surrounding environment and changes in lifestyles [9,10,11,12,13,14,15,16,17].

In this study, we aimed to, firstly, compare the lipid profiles of 4 different insulin-sensitive tissues in lean individuals. Secondly, and most importantly, we wanted to investigate the plasticity of each tissue lipidome and its capacity to recover when individuals are subjected to opposing and sequential lifestyle routines: high-fat feeding (HFD group), followed by a switch to a healthy regime with changes in nutritional intake and physical exercise (INT group).

In the reference state (lean healthy mice), we found that each tissue exhibits a unique lipid profile and a tissue-specific lipid distribution (Appendix A) that is in agreement with previous findings [6,8,18,19,20]. Other lipidomic studies looked into the effect of changes in lifestyle [9,10,11,12,13,14,15,16,17] focusing mainly on the effects of obesity in the lipidome of one particular tissue, or on changes in content of a specific category of lipid species, such as glycerolipids, in several tissues. In our study, however, we aimed to assess the lipidome of 4 tissues from all 3 groups (Ctrl, HFD and INT). Hypothalamus showed no significant modifications in its lipid profile amongst the 3 groups, while gastrocnemius, liver and eWAT demonstrated alterations in their lipid profiles in response to the interventions. Gastrocnemius and liver both exhibited a lipidome that is reflective of the energy balance and body composition. When a healthier lifestyle was adopted, the changes in the lipidome due to HFD were reversed. The liver showed pronounced changes among groups, which proves the role of this organ in lipid metabolism. On the other hand, eWAT disclosed a lipidome not as sensitive to energy balance but reflective of the lipid composition of the administered diet, emphasizing its role in the storage of lipids. For example, the UFA content in eWAT perfectly mimics the composition of both HFD and INT diets in UFAs (Table 2 and Figure 6).

TAG, a glycerolipid, is considered the main form of energy storage in the organism and its main site of storage is WAT, enriched at least ~4- to 30-fold as compared to any other tissue [6,8,20]. Our data reveal TAG content in eWAT to be at least ~18-fold higher than in liver or in gastrocnemius. DAG was not detected in gastrocnemius and free glycerol was only detected in eWAT. While other studies have shown the presence of glycerolipids in the lipidome of the hypothalamus [21,22], they remained undetected in our animals, potentially due to the sensitivity of the technique. On the other hand, sterol lipids were highly abundant in hypothalamus, as expected for the high density of cell membranes in the brain [23]. The exception was esterified cholesterols, found to be enriched in liver. Cholesterol homeostasis is maintained by the liver; through the enzyme Acyl-CoA cholesterol acyltransferase *ACAT2*, free cholesterol is esterified into cholesterol esters that are incorporated into lipoproteins [24].

Fatty acids are derived mainly from TAGs, phospholipids and cholesterol esters. Thus, with the exceptionally high content of TAG in eWAT, it was expected that eWAT would account for the highest FA content, as it was observed. Liver and hypothalamus were also enriched in FA, with the liver being a major tissue in lipid metabolism (synthesis and storage) and the hypothalamus having a high cell density, and hence a high content in structural lipids such as phospholipids and cholesterol esters. Gastrocnemius showed the lowest content in FA. Similarly, PLs content reflected tissue structure by exhibiting the lowest content in gastrocnemius and the highest in hypothalamus. It is worth mentioning that in mammalian cells, PC and PE are the most abundant phospholipids, comprising 60–80% of the total [6,8,19,25]. This is reflected in our data, that also revealed PE to be more abundant than PC.

Due to the complexity of the changes on the distribution and content in each tissue upon the nutritional and lifestyle interventions observed in our test groups (Ctrl, HFD and INT), they will be discussed independently. In our model, there was no response in the hypothalamus to any of the interventions, which is in contradiction to previous studies. After 6–8 weeks under similar dietary conditions to our model, HFD induced a 2–3-fold increase in most classes of lipids in the hypothalamus in one study [22], the levels of PC, TAG, DAG and some ceramides important for signalling were mostly affected in another [21]. Assessing the lipidome in different models of high-fat feeding, Borg et al. demonstrated that hypothalamic lipid accumulation is not driven by obesity but by diets enriched in fatty acids [21]. Several neurological disorders are characterised by an increase in levels of certain lipid species; hence, being at risk for developing such disorders increases with higher prevalence of obesity. For example, in Alzheimer’s disease, HIV, arteriosclerosis and ageing, long-chain ceramides are found to be increased [21]. In an attempt to reverse the lipid accumulation detected in the hypothalamus after HFD feeding, an exercise training in parallel with HFD was proven inefficient [21], while a return to a low-fat diet for a similar period of time reversed the effect of HFD on the lipid profile [22]. Our data do not support these observations, although the differences could be due to limitations related to the analytical tool utilised in our report (NMR) as compared to the other studies based on mass spectrometry (MS). NMR, in most cases, is more quantitative than MS but less sensitive. The major difference between these two techniques is that NMR gives quantitative information about a lipid family as a whole, while MS detects the abundance of individual lipid species, being more sensitive (detecting differences in carbon chain length and degree of unsaturation). Thus, the difference between techniques could explain the diversity of results in the literature.

Another important tissue in the regulations of carbohydrate and lipid metabolism in the organism is skeletal muscle. IR in skeletal muscle was initially related to total content of intramyocellular lipids (IMCL). But more recent studies have proved that particular lipid intermediates are better indicators for skeletal muscle IR than total IMCL [26,27,28,29,30,31,32]. Observations of decreased glycogen synthesis, insulin-stimulated *IRS-1* tyrosine phosphorylation and *PI3K* activity in skeletal muscle upon an acute increase in plasma FAs suggest that FAs inhibit insulin-stimulated glucose uptake [26,27,28]. It has been suggested that ceramides and DAG are the main lipid intermediates inducing skeletal IR [33,34]. In the exception of the “Athlete’s paradox” case [34], increased intramyocellular triacylglycerols (IMTG) have been associated with IR even though IMTG are considered metabolically inert and do not directly interfere with proximal insulin signalling [35]. It is suggested that the expanded IMTG pools generate the lipid intermediates that participate in the development of IR [35]. In our model, gastrocnemius lipidomics revealed a 3-fold increase in IMTG in the HFD group. Another significant change in this group was an increase in content of FAs: total UFA, but mainly oleic and linoleic acids. An increase in FA content could be explained by an increase in the ratio between lipid synthesis and fatty acid oxidation. Some investigations revealed a compromise in the ability of mitochondria to oxidise FA [36,37,38], decreases in mitochondrial content but no intrinsic alterations, or dysfunction, within the mitochondria [39], or increased mitochondrial content and fatty acid oxidation capacity [40,41,42]. Based on our data, the most plausible explanation for increased FA content is the uptake of FA in excess, due to overnutrition and WAT insulin resistance (ectopic lipid accumulation) that could not be counterbalanced by improved mitochondrial oxidative capacity. Interventions, such as changes in diet and exercise, have proven to reverse this effect, as clearly observed in our study in the INT group in IMTGs and FAs regressing to Ctrl levels. It is well-documented that during exercise, muscle fibres oxidise FAs from IMTGs and mobilise IMCLs stored in lipid droplets between sarcomeres to form an energy pool [31,43,44,45]. In our study however, FA content comprises not only the levels of FFAs but also esterified lipids. Thus, even though there were no changes in the content of cholesterols and PLs among groups, the FA composition of these lipid species might have been altered. Major changes in the composition of the FA profile in skeletal muscle induced by changes in diet, especially the administration of n-3 long chain poly-unsaturated fatty acids (LCPUFAs), have been observed, with minor changes after exercise [46,47]. Consistently, the INT group showed high levels of n3-LCPUFAs, in line with a study that demonstrated increased n-3 LCPUFAs in membranes when substituting dietary linoleic acid with α-linolenic acid (refer to Table 2) [48]. Increased levels of n-3 LCPUFAs have been positively correlated with improved insulin action in muscle [47]. Collectively, our results on the skeletal muscle lipidome support the notion that on an energy surplus, both IMTGs storage and FA content increases, while changes in dietary contents of FAs promote the accumulation of n-3 PUFAs.

The liver is the main site of FA synthesis and lipoprotein assembly. FAs are used in the liver mainly in three pathways: esterification into TAGs that are stored in lipid droplets, excretion in lipoproteins such as very-low-density lipoprotein (VLDL) and β-oxidation in liver mitochondria [49]. TAGs are the most common energy compounds of FAs for storage and circulation [50]. Thus, with overnutrition, hepatic accumulation of TAGs and hepatic FA content are increased [16]. Also, liver is responsible for the intracellular esterification of free cholesterol (FC) through the enzyme *ACAT2*, preventing an excess of toxic FC in cells [16,24]. Pathological accumulation of lipid droplets in the liver results in non-alcoholic fatty liver disease (NAFLD), a common feature of obesity [49,51]. In our study, livers from HFD animals showed a 6-fold increase in TAGs as well as increased levels of FAs, esterified cholesterols and total cholesterol. Changes in liver phospholipids were reflected by a decrease in PE abundance, which is in accordance with a previous study using a lard-based HFD (refer to Table 1) [52]. Synthesis of PE in liver mitochondria is reliant on the activity of mitofusin-2, which is found to be downregulated in HFD-fed individuals [53]. Although the dynamics of lipid droplets in liver are not fully understood, a decrease in hepatic lipid content stimulated by exercise has been shown, even without changes in body weight [43,49,54]. In contrast, a new study has shown a reduction in liver weight in the HFD group following an exercise program, accompanied by reduced lipid droplets, but no reduction in intrahepatic TAGs [51]. Several studies have reported a beneficial effect of exercise on hepatic fatty acid oxidation. Thus, exercise has been shown to increase the activation of rate-limiting enzymes such as carnitine palmitoyl-CoA transferase I/II (*CPT-1* and *CPT-2*), acyl-coenzyme A dehydrogenase (*ACD*), and decrease the expression of other lipogenic enzymes such as acetyl-CoA carboxylase (*ACC*), fatty acid synthase (*FAS*), elongases and stearoyl-CoA desaturase 1 (*SCD1*) [49]. In our model, the INT group showed a regression to Ctrl levels in the hepatic content of TAGs, FAs and cholesterols. Taking into account the present findings, the composition of the diet and the daily caloric intake of the HFD and INT groups (Table 1 and Table 3), it can be proposed that the liver content in glycerophospholipids, FAs and sterols is highly reflective of the energy balance. In addition, the liver lipidome showed a high capacity for recovery as most of the changes induced by HFD were reversed.

WAT is nowadays considered a highly regulated energy storage site and a major player in energy homeostasis [55]. WAT stores energy as TAGs and releases fatty acids via lipolysis for other organs to use. Under energy surplus, WAT undergoes a process of tissue remodelling and functional alterations characterised mostly by adipocyte hypertrophy, low-grade inflammation, local hypoxia and modification in adipokine release [55,56]. When the capacity of WAT to adapt and store this surplus of energy proportionally becomes limited, the excess FAs are deposited in non-adipose sites, mainly in the liver, skeletal muscle, heart and kidneys [2,57,58,59]. The extent of ectopic fat accumulation is significantly linked to the ability of visceral AT to expand [2,57,60]. eWAT shows a defect in insulin-stimulated glucose uptake at a very early time point after a HFD when peripheral tissues remained insulin-sensitive [20]. When HFD feeding is prolonged, TAG and DAG levels in eWAT were elevated as well as ceramide and sphingomyelin in mice models [20] and rat models [61]. Analysis of the lipid profile of eWAT in the present study showed an increase in both total and free cholesterol levels. Several studies have suggested that dietary fat composition could define the effect of a HFD on the pathophysiology of obesity; SFAs, in comparison to UFAs, worsen the inflammatory profile in eWAT and make it less reversible in eWAT and systemically [59,61,62]. To evaluate whether or not HFD and INT eWAT profiles mirrored the composition of the diets and the intake of each lipid, firstly, we showed that even though HFD and INT group diets have the same percentage of fat, INT mice consumed less SFAs and more UFAs (Table 1, Table 2 and Table 3). We discovered that the UFAs are highly mirrored in the eWAT lipidome of INT mice (Table 2), with a decrease in the levels of omega6 FAs (linoleic acid and ARA) and an increase in the levels of PUFA, linolenic acid and combined n3-DHA-EPA-linolenic. The highlight of our data on eWAT is the reflection of the exact composition of the administered diets on the lipidome of the tissue, which is not observed in any of the other 3 insulin-sensitive tissues analysed.

Many of our results could not be explained solely based on our hypotheses or previously published studies. Alterations in some lipid species could not be associated to only the direct effect of either the energy balance or the administered diet. For example, looking at the hepatic phospholipids content, a single pattern could not be defined, which means each of these lipid classes interacts differently in liver. On the other hand, PLs show a tendency to increase in eWAT, moving from Ctrl to HFD to INT, that might be associated to the expandability and remodelling of the WAT. We can associate the alterations in content in phospholipids and SM, which are main components of cellular membranes, to changes in the cellularity of the tissue. Based on previous findings from our laboratory, the HFD stimulated the expansion of WAT through hypertrophy and hyperplasia, while the eWAT of the INT group showed a pronounced infiltration of immune cells and small adipocytes sign of tissue remodelling [63].

Thus, the modifications in lipidomes due to different environmental cues cause specific responses, which we do not fully understand yet. In this regard, it is well-known that mitochondria play a role in lipid metabolism [3]. Previous data from our laboratory [63] on mitochondrial function in several tissues reveals no changes in mitochondrial respiration in hypothalamus, glycolytic skeletal muscles and liver, tissues that demonstrate a reversible lipidome after HFD and INT. In contrast, eWAT showed a significant decrease in mitochondrial function in the HFD group which was maintained even after INT. Mitochondrial dysfunction detected in eWAT in a previous study with similar experimental design [63] could explain the lack of recovery of the eWAT lipidome after HFD. Therefore, further studies are required to better understand the dynamics of the different lipid classes in relation to mitochondria and other organelles, such as peroxisomes, endoplasmic reticulum (ER) and lipid transport and recycling.

Based on the data described above, we may come to several conclusions. We were able to establish that the lipid profile of three insulin-sensitive tissues revealed major alterations stimulated by changes in lifestyles in a tissue-specific manner. The lipidomes of gastrocnemius and liver are more susceptible to changes in energy surplus. This capacity to adapt to different dietary interventions is an indication that both tissues maintain their plasticity. On the other hand, eWAT lipidome mimicked the content of the administered diet and not the energy balance, which was made most clear by the data of its fatty acids content. Finally, we cannot forget the potential effect of these two factors, energy balance and diet composition, on key cellular components and their plasticity. When they become dysfunctional, ER, peroxisomes and mitochondria, that are critical for lipid metabolism, induce further alterations to the lipidome. Before a potential translation of our findings to human and clinical practice, it should be determined whether the lipidomic profiles are species-specific.

## 4. Materials and Methods

### 4.1. Animals

Male *C57BL/6JOlaHSD* mice were purchased from Envigo (Indiana, IN, USA) at 5 weeks of age. They were split into two groups and fed *ad libitum* with different diets for 16 weeks: control group (Ctrl) fed with chow diet from Teklad Global 14% Protein Rodent Maintenance Diet by Envigo, and a high-fat diet-fed group (HFD) fed with the diet D12451 by Research Diets, New Brunswick, NJ, USA. After 16 weeks on HFD, the criteria for inclusion of mice in the HFD group were overweight, hyperglycaemia and hyperinsulinemia. Once this pathological condition was achieved, some of the HFD mice were randomly chosen to undergo a lifestyle intervention, thus the third group, INT, was created. Specifically, the INT group was put under a calorie restriction and a change of diet regime for 5 weeks. INT mice were fed with intervention diet (Preliminary Formula Rodent Diet with 45 kcal% Fat and Modification with Flaxseed and Olive Oil by Research Diets, New Brunswick) (Table 1 and Table 2). Also, an exercise program was imposed on INT mice (as detailed below). All animal procedures were approved by the local ethics committee, Comitè Ètic d’Experimentació Animal at the Universitat de Barcelona and the Departament d’Agricultura, Ramaderia, Pesca, Alimentació i Medi Natural at the Generalitat de Catalunya, complying with current Spanish and European legislation.

### 4.2. Nutritional and Exercise Intervention

For 5 weeks, INT mice were under a nutritional and exercise intervention. Their daily energy intake was 80% in kcal of the daily energy intake determined for a Ctrl animal for the first week and then 100% for the remaining 4 weeks. The intervention diet had the same energy content from fat (45%) as the HFD but included several nutritional modifications: sucrose was replaced by corn starch and fat sources were changed from lard and soybean oil to flaxseed oil and olive oil (Table 1). The reason behind the alterations in fat sources was to decrease SFA content and increase UFA (Table 2). The intervention also included an exercise program where mice were allocated to treadmill exercise (Exer-6M Open Treadmill for Mice and Rats with Shocker and Software 2–102 M/m Columbus Instruments; Columbus, OH, USA) during 1 h per day/5 days per week. The protocol for the exercise was as followed: acclimatisation during the first week of intervention followed by a gradual increase in speed to reach 20 m/min, with a second level inclination (10°) by the end of the 5-week period.

### 4.3. Glucose Homeostasis In Vivo Functional Assay

Intraperitoneal glucose tolerance test (IGTT) was performed after 16 h fasting. D-Glucose at 2 g/kg of mouse body weight (BW) was administered at time 0. At time points 15, 30, 60 and 120 min after glucose administration, blood glucose levels were measured and recorded using a glucometer.

### 4.4. Indirect Calorimetry

Mice from each experimental group were used as subjects in the TSE LabMaster (TSE Systems, Bad Homburg, Germany), as described previously [63]. During the first 24 h, the mice were acclimated and then monitored for 48 h. Every 30 min, data of O_2_ consumption and CO_2_ production were collected. The calorimeter system provided a standard analysis software allowing the determination of energy expenditure using the collected data. Heat production was calculated using the abbreviated Weir equation ([3.94(VO_2_) + 1.11(VCO_2_)] 1.44). Other measured parameters during the 48 h were food and water intake and locomotor activity. Thirty min respiratory exchange ratio (RER) values were averaged for each animal at the same time point across 24 h. The median value for each animal was considered baseline, and the deviation to it was calculated, along with the maximal and minimum values.

### 4.5. Sample Preparation for NMR Metabolomics

Liver, eWAT, gastrocnemius and hypothalamus were removed and immediately placed in liquid nitrogen. Using mortar and pestle, the frozen tissues were pulverised. For each of eWAT, gastrocnemius and hypothalamus, 30–70, 20–40 and 10–25 mg of powder respectively, were mixed with methanol and ultrasonicated. Chloroform was added in 2 steps to a final 1:1 concentration and then water was added to get 1:2:2 (water:methanol:chloroform). After centrifugation, the upper aqueous phase (methanol/water) was separated from the lower organic phase and both were collected and stored. For the liver samples, 50–100 mg of liver powder was mixed with ice-cold acetonitrile:water (1:1), ultrasonicated, centrifuged and the aqueous upper phase was collected. After 3 repeats, the collected and combined aqueous supernatant was frozen and lyophilised. The resultant pellet was dried and extracted with chloroform:methanol (2:1) by ultrasonication and centrifuged. The upper lipidic phase was collected and dried under N2 stream. For the measurement experiment, the organic extract was reconstituted in 700 uL of CDCl_3_/CD_3_OD (2:1) solution (containing tetramethylsilane, TMS). The supernatants from the newly formed solution were transferred into 5 mm NMR tubes. The different protocols for sample preparation between liver and the other tissues, mainly the use of acetonitrile (liver), could affect the detection of some polar lipids. However, the detected metabolites in liver could be compared to the other tissues.

### 4.6. Nuclear Magnetic Resonance (NMR) Metabolomics Analysis

^1^H-NMR spectra were recorded at 310 K on a Bruker Avance III 600 spectrometer (Bruker, Billerica, USA) operating at a proton frequency of 600.20 MHz using a 5 mm TCI CryoProbe triple resonance (^1^H, ^13^C, ^31^P). A 90° pulse with water pre-saturation sequence (zgpr) was used. Thus, a 50 Hz power irradiation was used during recycling delay and mixing time to pre-saturate the solvent. 256 transients were collected into 64 k data points for each ^1^H spectrum, with the spectral width set as 12 kHz (20 ppm). The exponential line broadening applied before Fourier transformation was of 0.3 Hz. The frequency domain spectra were manually phased and baseline-corrected using TopSpin software (version 2.1, Bruker, Billerica, USA).

### 4.7. NMR Data Analysis

^1^H-NMR data acquisition was performed as previously described [63]. Metabolite identification was performed using lipid standards and according to Reference [64]. After baseline correction, specific ^1^H-NMR regions identified in the spectra were integrated using the AMIX 3.8 software package. Then, each integration region was normalised by proton number and then, by the tissue weight used from each sample. Data (pre-)processing, data analysis and statistical calculations were performed in RStudio (R version 3.0.2).

### 4.8. Statistical Analysis

Results are expressed as mean ± SEM. The statistical significance among the three experimental groups was assessed using one-way analysis of variance (ANOVA), and differences between means were subsequently tested by Tukey’s post-hoc test. A *p*-value < 0.05 was considered significant in all cases, meaning a confidence interval of 95% and setting significance level at α = 0.05. Tendencies with *p*-value between 0.05 and 0.07 are also indicated. All statistical analyses were performed using GraphPad Prism 6 (GraphPad Software, Inc. La Jolla, CA, USA).

## Figures and Tables

**Figure 1 ijms-22-03694-f001:**
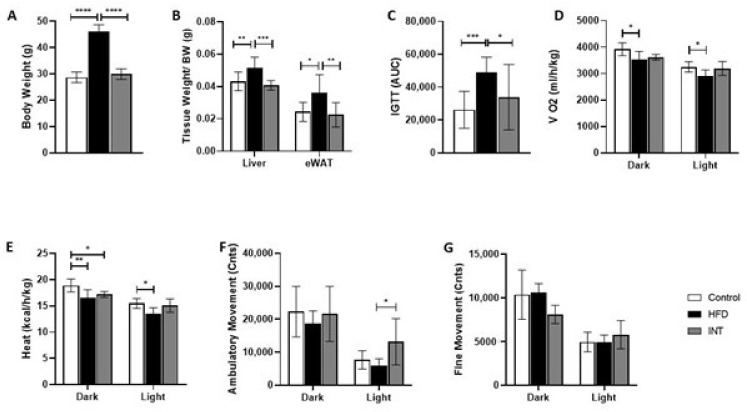
Phenotype overview. (**A**) Body weight (BW) (control (Ctrl) *n* = 13; high-fat diet (HFD) *n* = 12; intervention (INT) *n* = 14). (**B**) Tissue weights of liver and epididymal white adipose tissue (eWAT) normalised to body weight (Ctrl *n* = 9; HFD *n* = 11; INT *n* = 10). (**C**) Glucose homeostasis, intraperitoneal glucose tolerance test (IGTT) after overnight (16 h) fasting represented as area under the curve (AUC) (Ctrl *n* = 13; HFD *n* = 13; INT *n* = 14). (**D**) Oxygen consumption. Average O_2_ volume (VO_2_) measurements normalised by BW during the dark and light phase of the day (Ctrl *n* = 10; HFD *n* = 5; INT *n* = 6). (**E**) Heat production normalised by BW during the dark and light phases (Ctrl *n* = 10; HFD *n* = 5; INT *n* = 6). (**F**) Activity represented by counts of ambulatory movement (Ctrl *n* = 10; HFD *n* = 5; INT *n* = 6). (**G**) Activity represented by counts of fine movement (Ctrl *n* = 10; HFD *n* = 5; INT *n* = 6). Data represented as mean ± SEM, one-way analysis of variance (ANOVA) and post-hoc Tukey’s tests, * *p* < 0.05, ** *p* < 0.01, *** *p* < 0.001, and **** *p* < 0.0001.

**Figure 2 ijms-22-03694-f002:**
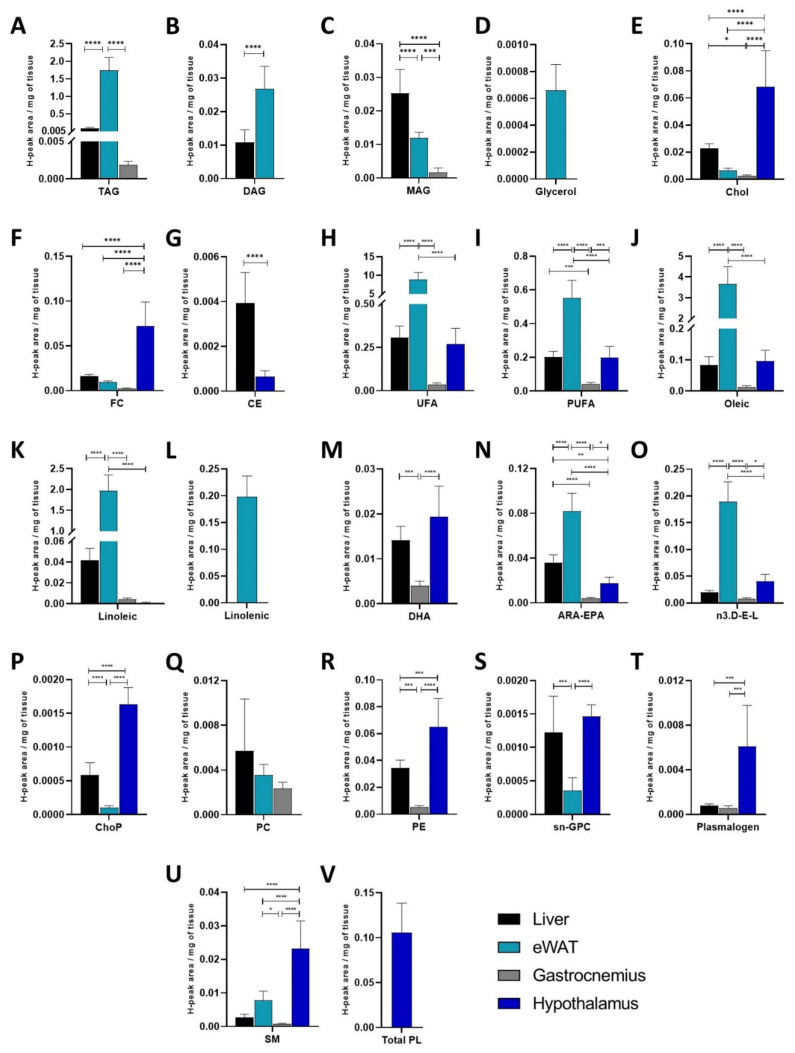
Lipid profiles of liver, eWAT, gastrocnemius and hypothalamus. Comparison of the presence of glycerolipids (**A**–**D**), sterol lipids (**E**–**G**), fatty acids (**H**–**O**), phospholipids and sphingolipids (**P**–**V**) in 4 different tissues from control mice. Liver *n* = 8, eWAT *n* = 8, gastrocnemius *n* = 8, hypothalamus *n* = 7. Y-axis values indicate proton-based NMR peak area normalised to sample weight (H-peak area/mg of tissue). Data represented as mean ± SEM, one-way ANOVA and post-hoc Tukey’s tests, * *p* < 0.05, ** *p* < 0.01, *** *p* < 0.001, and **** *p* < 0.0001.

**Figure 3 ijms-22-03694-f003:**
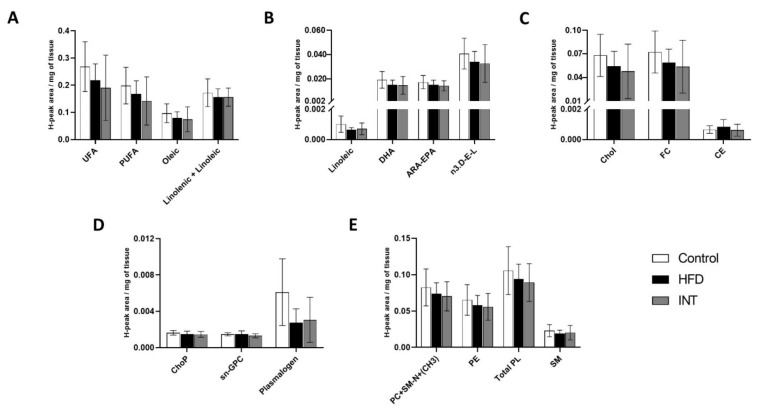
Alterations in metabolite lipid profile in hypothalamus. Representation of the changes induced by HFD and the nutritional and exercise intervention following the HFD, on the presence of (**A**,**B**) fatty acids, (**C**) sterol lipids and (**D**,**E**) phospholipids and sphingolipids. Ctrl *n* = 7, HFD *n* = 8, INT *n* = 8. Y-axis values indicate H-peak area/mg of tissue. Data represented as mean ± SEM, one-way ANOVA and post-hoc Tukey’s tests.

**Figure 4 ijms-22-03694-f004:**
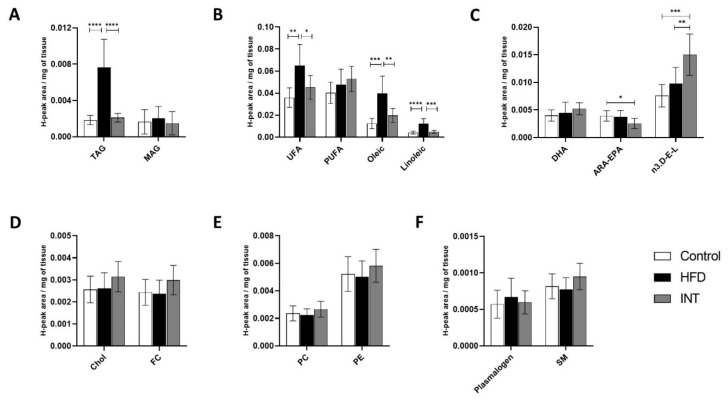
Alterations in metabolite lipid profile in gastrocnemius. Representation of the changes induced by HFD and the intervention following the HFD on the presence of (**A**) glycerolipids, (**B**,**C**) fatty acids, (**D**) sterol lipids and (**E**,**F**) phospholipids and sphingolipids. Ctrl *n* = 8, HFD *n* = 7, INT *n* = 7. Y-axis values indicate H-peak area/mg of tissue. Data represented as mean ± SEM, one-way ANOVA and post-hoc Tukey’s tests, * *p* < 0.05, ** *p* < 0.01, *** *p* < 0.001, and **** *p* < 0.0001.

**Figure 5 ijms-22-03694-f005:**
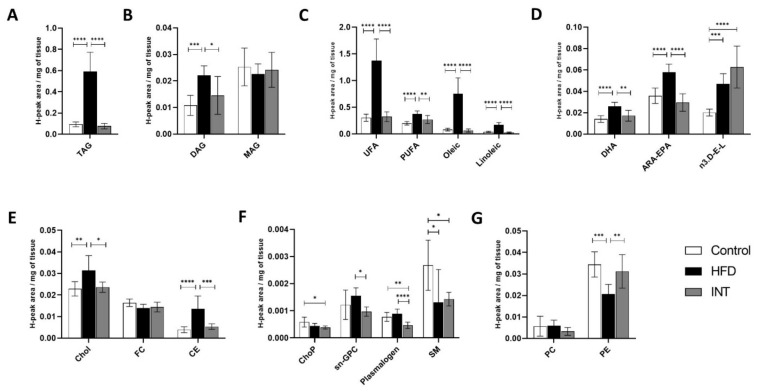
Alterations in metabolite lipid profile in liver. Representation of the changes induced by HFD and the intervention following the HFD on the presence of (**A**,**B**) glycerolipids, (**C**,**D**) fatty acids, (**E**) sterol lipids and (**F**,**G**) phospholipids and sphingolipids. Ctrl *n* = 8, HFD *n* = 8, INT *n* = 7. Y-axis values indicate H-peak area/mg of tissue. Data represented as mean ± SEM, one-way ANOVA and post-hoc Tukey’s tests, * *p* < 0.05, ** *p* < 0.01, *** *p* < 0.001, and **** *p* < 0.0001.

**Figure 6 ijms-22-03694-f006:**
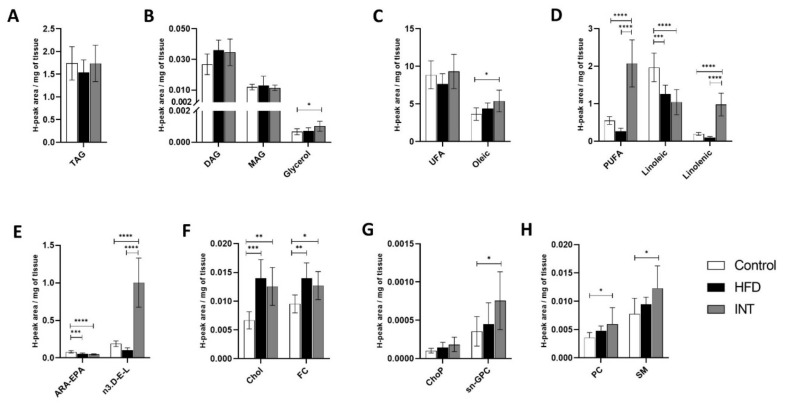
Alterations in metabolite lipid profile in eWAT. Representation of the changes induced by HFD and the intervention following the HFD on the presence of (**A**,**B**) glycerolipids, (**C**–**E**) fatty acids, (**F**) sterol lipids and (**G**,**H**) phospholipids and sphingolipids. Ctrl *n* = 8, HFD *n* = 8, INT *n* = 7. Y-axis values indicate H-peak area/mg of tissue. Data represented as mean ± SEM, one-way ANOVA and post-hoc Tukey’s tests, * *p* < 0.05, ** *p* < 0.01, *** *p* < 0.001, and **** *p* < 0.0001.

**Table 1 ijms-22-03694-t001:** Nutritional composition of the different diets used. Comparison in the energy content, macronutrients source and distribution of the 3 different diets.

		Chow Diet	High-Fat Diet	Intervention Diet
Caloric Density (kcal/g)		2.9	4.7	4.7
Protein (%)		20	20	20
Carbohydrates (%)		67	35	35
Fat (%)		13	45	45
Total carbohydrates				
Source (%)	Corn starch		21.1	63.8
	Maltodextrin		28.9	36.2
	Sucrose		50	-
Total fat				
Source (%)	Soybean oil		12.3	-
	Lard		87.7	-
	Flaxseed oil		-	55.6
	Olive oil		-	44.4
Composition (%)	Saturated fat	17.6	31.7	11.8
	Monounsaturated fat	20.6	35.6	43.8
	Polyunsaturated fat	61.8	32.7	44.4

**Table 2 ijms-22-03694-t002:** Fatty acids composition of HFD and INT diet and the daily fatty acids intake of the different experimental groups. Distribution of saturated and unsaturated fatty acids in HFD and INT diets. Daily intake of HFD and INT mice in saturated fatty acid (SFA) and unsaturated fatty acids (UFA).

	High-Fat Diet		Intervention Diet	
	Sources (g)	Daily Intake (mg)	Sources (g)	Daily Intake (mg)
Total fat	202.5	600	202.5	450
C16, Palmitic	36.8	109.0	14.9	33.1
C18, Stearic	19.8	58.7	5.6	12.4
C18:1, Oleic	64.1	189.9	83.7	186.0
C18:2, Linoleic	56.2	166.5	29.7	66.0
C18:3, Linolenic (ALA)	4.2	12.4	62.4	138.7
C20:4, Arachidonic (ARA)	0.5	1.5	0	0
C20:5, Eicosapentaenoic (EPA)	0	0	0	0
C22:6, Docosahexaenoic (DHA)	0	0	0	0

**Table 3 ijms-22-03694-t003:** Daily caloric and macronutrient intake of the different experimental groups.

	Ctrl	HFD	INT
Daily Intake (g)	3.08	2.55	1.9
Caloric Density (kcal/g)	2.90	4.73	4.73
Daily Caloric Intake (kcal)	8.93	12.06	8.99
	Daily Intake of Macronutrients (g)		
Protein	0.45	0.60	0.45
Carbohydrate	1.50	1.06	0.79
Fat	0.13	0.60	0.45

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
