# Peer review of "Effects of Lifestyle Intervention in Tissue-Specific Lipidomic Profile of Formerly Obese Mice"

_ijms, 2021, doi:10.3390/ijms22073694_

Round 1

Reviewer 1 Report

Summary

Dahdah et al. showed the impact of high fat diet (HFD) and an intervention diet (INT) on the lipidome of multiple mouse tissues types responsive to insulin signaling by 1H-NMR. With the intervention diet and additional physical exercise the observed changes induced by HFD could be partially reversed. The investigate tissues also reacted in different ways, for instance hypothalamus did not show any lipidome changes when the three studied conditions (control, HFD, INT) were compared. NMR is a suitable technology to detected differences in the lipidome based on lipid classes.

Broad comments

Only male mice were subjected to the three diets. How would female mice react to it and what is the rational behind just choosing male mice?

1H-NMR was used to observe changes in the lipidome. Even though tetrametylsilane was added to the prepared lipid extracts it was not used for quantification. Instead of having arbitrary units, which are quite difficult to compare to other studies, in my opinion it would make sense to quantify individual lipid classes and present their absolute amounts normalized to the tissue weight used for extraction. Another issue is that liver was extracted completely in a different way compared to the other tissue types. How can the data then be compared? For instance, ACN will remove to some extend polar lipids.

Specific comments

Line 158: Somehow it is rather odd that total phospholipids were only detected in hypothalamus. Can you elaborate?

Figure 2: it should be clearly stated that the presented values are coming from the control condition.

Line 175-179: Based on the available data it is difficult to explain that hypothalamus is not affected by the used diets. It should be stated that NMR gives only information on the overall lipid class level, but not on its composition and potentially hypothalamus is reacting to the diets, by changing its lipid composition e.g. total carbon length and level of unsaturation.

Line 219: the abbreviation of SPME is introduced but not explained.

Figure 5: it is absolutely necessary to specify what the difference between cholesterol and free cholesterol is. The explanation should be stated as early as possible to avoid any confusion.

Line 336: with a proper method MS can be as quantitative as NMR. Please rephrase.

Line 338: As indicated above MS can quantify lipid species as well as lipid classes by summing up lipid species belonging to this class. Please rephrase.

Line 440: rather lipid classes then lipid species.

Line 457: as above.

Line 528-532: is there any reason why liver was extracted in a quite different way compared to the other tissue types. Also ACN will remove more polar lipids to some extend. How can this tissue be compared to the others when the preparation is completely different?

Statistical analysis: what’s the n number per condition? Was the data tested for normality before subjecting it to one-way Anova and Tukey post-hoc test?

Reviewer 2 Report

Title: Effects of lifestyle intervention in tissue-specific lipidomic profile of formerly obese mice.

Authors: Norma Dahdah, Alba Gonzalez-Franquesa, Sara Samino, Pau Gama-Perez, Laura Herrero, José Carlos Perales, Oscar Yanes, Maria Del Mar Malagón and Pablo M García-Rovés

General comment:

Behavioral interventions play a central role in the management of obesity. However, there is also a therapeutic dilemma about what type of intervention should be chosen for the individual to achieve optimal health benefits. This is since data on the influence of particular lifestyle modifications on the structure and function of critical organs involved in the pathogenesis of obesity-related complications is still insufficient. In their work, Norma Dahdah et al. evaluated the effects of a dual body-weight reducing intervention (diet and physical activity) on tissue-specific lipidomic profile of formerly obese mice. The research hypothesis is innovative, the methodology was adequately chosen, and the results are valuable for clinical practice. Therefore I have only minor remarks regarding the manuscript.

Minor revisions:

  1. Discussion – some limitations of the study should be discussed, e.g., the fact that the observed alternations in lipidomic profiles can be species-specific. Subsequently, some directions for further research could be suggested.
  2. Figure 2 – please indicate in the leged that the results concern the control group.
  3. Supplementary Figure 1 – please indicate which studied group the figure concerns
  4. Please explain the abbreviations as they appear in the text, e.g., HFD, AUC, eWAT, IGTT, ACAT.

Author Response

General comment:

Behavioral interventions play a central role in the management of obesity. However, there is also a therapeutic dilemma about what type of intervention should be chosen for the individual to achieve optimal health benefits. This is since data on the influence of particular lifestyle modifications on the structure and function of critical organs involved in the pathogenesis of obesity-related complications is still insufficient. In their work, Norma Dahdah et al. evaluated the effects of a dual body weight reducing intervention (diet and physical activity) on tissue-specific lipidomic profile of formerly obese mice. The research hypothesis is innovative, the methodology was adequately chosen, and the results are valuable for clinical practice. Therefore, I have only minor remarks regarding the manuscript.

We want to thank the reviewer for the comments and suggestions to improve the quality of the manuscript. We agree that the results presented in this study are of potential interest for clinical practice and as suggested by the reviewer we have stated in the Discussion the limitations in relation to its translational potential.  

Minor revisions:

  1. Discussion – some limitations of the study should be discussed, e.g., the fact that the observed alternations in lipidomic profiles can be species-specific. Subsequently, some directions for further research could be suggested.

Answer to reviewer 2:

We have added the following sentence at the end of the Discussion considering this comment.

“Before a potential translation of our findings to human and clinical practice it should be determined whether the lipidomic profiles are species-specific”.

  1. Figure 2 – please indicate in the legend that the results concern the control group.

Answer to reviewer 2:

Done

  1. Supplementary Figure 1 – please indicate which studied group the figure concerns.

Answer to reviewer 2:

Done

  1. Please explain the abbreviations as they appear in the text, e.g., HFD, AUC, eWAT, IGTT, ACAT.

Answer to reviewer 2:

            Thanks for pointing this out. Done